# Effects of the Interaction between Rumen Microbiota Density–VFAs–Hepatic Gluconeogenesis on the Adaptability of Tibetan Sheep to Plateau

**DOI:** 10.3390/ijms25126726

**Published:** 2024-06-19

**Authors:** Wenxin Yang, Yuzhu Sha, Xiaowei Chen, Xiu Liu, Fanxiong Wang, Jiqing Wang, Pengyang Shao, Qianling Chen, Min Gao, Wei Huang

**Affiliations:** Gansu Key Laboratory of Herbivorous Animal Biotechnology, College of Animal Science and Technology, Gansu Agricultural University, Lanzhou 730070, China; aaaaa0108@163.com (W.Y.); shayz@st.gsau.edu.cn (Y.S.); cxw20002022@163.com (X.C.); m19893318751@163.com (F.W.); wangjq@gsau.edu.cn (J.W.); shaopengyang666@163.com (P.S.); chenqianling223@163.com (Q.C.); gm12017101@163.com (M.G.); 18294737108@163.com (W.H.)

**Keywords:** Tibetan sheep, rumen microbiota density, VFAs, transport genes, gluconeogenesis

## Abstract

During the adaptive evolution of animals, the host and its gut microbiota co-adapt to different elevations. Currently, there are few reports on the rumen microbiota–hepato-intestinal axis of Tibetan sheep at different altitudes. Therefore, the purpose of this study was to explore the regulatory effect of rumen microorganism–volatile fatty acids (VFAs)–VFAs transporter gene interactions on the key enzymes and genes related to gluconeogenesis in Tibetan sheep. The rumen fermentation parameters, rumen microbial densities, liver gluconeogenesis activity and related genes were determined and analyzed using gas chromatography, RT-qPCR and other research methods. Correlation analysis revealed a reciprocal relationship among rumen microflora–VFAs-hepatic gluconeogenesis in Tibetan sheep at different altitudes. Among the microbiota, *Ruminococcus flavefaciens* (*R. flavefaciens*), *Ruminococcus albus* (*R. albus*), *Fibrobactersuccinogenes* and *Ruminobacter amylophilus* (*R. amylophilus*) were significantly correlated with propionic acid (*p* < 0.05), while propionic acid was significantly correlated with the transport genes monocarboxylate transporter 4 (*MCT4*) and anion exchanger 2 (*AE2*) (*p* < 0.05). Propionic acid was significantly correlated with key enzymes such as pyruvate carboxylase, phosphoenolpyruvic acid carboxylase and glucose (Glu) in the gluconeogenesis pathway (*p* < 0.05). Additionally, the expressions of these genes were significantly correlated with those of the related genes, namely, forkhead box protein O1 (*FOXO1*) and mitochondrial phosphoenolpyruvate carboxykinase 2 (*PCK2*) (*p* < 0.05). The results showed that rumen microbiota densities differed at different altitudes, and the metabolically produced VFA contents differed, which led to adaptive changes in the key enzyme activities of gluconeogenesis and the expressions of related genes.

## 1. Introduction

Altitude is one of the major environmental factors that affects the survival of organisms, and organisms evolve specific phenotypic survival traits in response to their environment [1]. Tibetan sheep bred on the Qinghai–Tibet Plateau, one of the oldest sheep breeds and one of the three major sheep breeds with coarse wool in China, are naturally raised throughout the year. After long-term evolution, they developed adaptive characteristics such as cold resistance, low oxygen tolerance, coarse feeding tolerance and strong disease resistance, providing local residents with meat, milk, fur and other living materials. Gradually, they have become a valuable livestock breed and a valuable biological genetic resource in the plateau region.

The rumen is a unique, dynamic and continuous digestive and metabolic organ of ruminants that serves as a “natural fermentation tank” for degrading fiber substances [2]. It is rich in various microorganisms such as bacteria, fungi and protozoa. The microbial community can be divided by function into cellulose-degrading bacteria, including *R. flavefaciens*, *R. albus* and *Fibrobacter subgenres*, *Treponema bryantii* (*T. bryantii*), *Butyrivibrio fibrisolvens* (*B. fibrisolvens*) and *Prevotella*; *Rumnobacter amycophilus* belonging to starch-degrading bacteria; *Clostridium butyricum* (*C. butyricum*) related to fat-degrading bacteria; and *Selenomonas ruminantium* (*S. ruminantium*), which originated from lactic acid-using bacteria [3,4]. They play important roles in forage fermentation and energy supply. Natural grazing is the main way to provide energy for Tibetan sheep, and natural grass is their main source of nutrition [5]. Due to differences in altitude, there are differences in environmental factors such as temperature, humidity and oxygen content in high-altitude areas, which in turn affect forage growth. Research has shown that with increasing altitude and decreasing aboveground biomass and height, grass exhibits a decrease in crude fiber content [6] and fiber-degrading bacteria, as the main microbial group producing VFAs leads to metabolic changes in VFA content. Therefore, after grazing, grasses experience anaerobic fermentation by rumen microorganisms, producing various VFAs, including acetic acid, propionic acid, butyric acid, isobutyric acid, valeric acid and isovaleric acid. Among them, propionic acid, acetic acid and butyric acid account for more than 95% of the total content [7]. The NH_3_-N in the fermentation products is also an important indicator that reflects the fermentation status and absorption rate of nitrogen-containing compounds in the rumen [8]. The rumen epithelium, as the main site of energy absorption, has an efficient VFA transport system, which is highly dependent on the surface area of the epithelium and the expressions of transport proteins [9]. The related transport proteins include monocarboxylate transporter carrier (*MCT*) [10,11], sodium hydrogen exchange protein (*NHE*) [12] and *AE2* [13]. Intestinal hepatic crosstalk has a profound impact on energy homeostasis. Compounds such as VFAs can enter the portal venous circulation and reach the liver to participate in glucose homeostasis. Among them, propionic acid is a major VFA that is involved in the activation of intestinal gluconeogenesis [14]. Studies have shown that propionic acid can increase the production rate of glucose in the liver. A significant supply of propionic acid not only increases the concentration of metabolites in the pyruvate and citric acid cycles of the liver but also accelerates endogenous glucose production [15]. Therefore, the amount of propionic acid absorbed by the portal vein increases, and the absorption and utilization of propionic acid by the liver significantly increase gluconeogenesis and glucose production through the up-regulation of transport genes. Therefore, VFAs can serve as signaling molecules that are transported by the rumen epithelium into the hepatic portal vein to participate in hepatic gluconeogenesis metabolism.

Glucose is an essential nutrient in animals’ lives and production activities. Approximately 80–90% of the glucose that is required by ruminant animals comes from the gluconeogenesis of propionic acid and glycogenic amino acids in the liver [16,17]. Gluconeogenesis is a continuous process involving multiple key enzymes, including pyruvate carboxylase (PC), phosphoenolpyruvic acid carboxylase (PEPCK), fructose-1,6-bisphosphonate (FBPase) and glucose-6-phosphatase (G6Pase) [18,19]. Adaptive evolution affects genes related to energy metabolism, helping ruminants live in high-altitude areas [20]. One study revealed that *FOXO1* can promote the expression of gluconeogenesis G6Pase and PEPCK [21]. The genes encoding *FOXO1*, peroxisome proliferator receptor gamma coactivator alpha (*PPARGC1A*), glucose 6-phosphatase catalytic subunit 1 (*G6PC1*) and *PCK2* [22], play key roles in determining the initiation of gluconeogenesis [13]. However, few studies have been conducted on rumen microbiota density–VFAs–hepatic gluconeogenesis interactions in plateau Tibetan sheep. Therefore, this study used the densities of the rumen microbiota in Tibetan sheep at different altitudes and the changes in VFAs as the starting points. By analyzing the associations with VFA-related transport proteins, the key enzymes involved in gluconeogenesis and related genes, the adaptive mechanism of the rumen microbiota in Tibetan sheep at different altitudes to liver gluconeogenesis was revealed. This study provides a new perspective on the adaptability of Tibetan sheep at different altitudes on the Qinghai–Tibet Plateau and a theoretical basis for it.

## 2. Results

### 2.1. Rumen Fermentation Parameters in Tibetan Sheep at Different Altitudes

The rumen fermentation parameters of Tibetan sheep living at different altitudes are shown in Table 1. The total amounts of VFAs at low altitudes were significantly greater than those at high altitudes (*p* < 0.001). Among them, the proportions of acetic acid, isobutyric acid, butyric acid and isovaleric acid were significantly higher at low altitudes than at high altitudes (*p* < 0.001), and the proportion of propionic acid was significantly greater at low altitudes than at high altitudes (*p* < 0.05), while the proportion of valeric acid was not significantly different between high and low altitudes (*p* > 0.05). The total NH_3_-N content was significantly greater at low altitudes than at high altitudes (*p* < 0.001).

### 2.2. Rumen Microbiota Density in Tibetan Sheep at Different Altitudes

The results of the relative density determinations of the ruminal flora of Tibetan sheep at different altitudes showed (Figure 1) that, except for *Prevotella*, which exhibited no significant difference (*p* > 0.05), the relative densities of *R. flavefaciens*, *R. albus*, *Fibrobacter succinogenes* (*F. succinogenes*), *S. ruminantium*, *T. bryantii*, *C. butyricum*, *R. amylophilus* and *B. fibrisolvens* at high altitudes were extremely high. At high altitudes, the *R. flavefaciens* density and *T. bryantii* density were significantly greater than those at low altitudes (*p* < 0.01); at low altitudes, the *B. fibrisolvens* density and *T. bryantii* density were greatest, with a 25.1-fold difference in expression; and at high and low altitudes, the *T. bryantii* density was the lowest.

### 2.3. Expression Levels of VFAs Transporter-Related Genes in the Rumen Epithelium of Tibetan Sheep at Different Altitudes

The results of VFA transporter-related gene expression measurements in the rumen epithelium of Tibetan sheep at different altitudes indicated (Figure 2) that the relative expressions of monocarboxylate transporter carrier 1 (*MCT1*) and *AE2* at high altitudes were very significantly greater than those at low altitudes (*p* < 0.01), and the relative expression of *MCT4* at high altitudes was significantly greater. At low altitudes (*p* < 0.05), there was no significant difference in the down-regulated in adenoma (*DRA*) and Na^+^/H^+^ exchange 3 (*NHE3*) genes between high and low altitudes (*p* > 0.05).

### 2.4. Gluconeogenic Enzyme Content and Related Gene Expression in the Liver of Tibetan Sheep at Different Altitudes

Analysis of the levels of key enzymes involved in gluconeogenesis and related gene expressions in the liver of Tibetan sheep at different altitudes (Figure 3) revealed that the FBPase concentrations in the liver of Tibetan sheep were the lowest and that the PC concentration was the highest. The PEPCK concentrations at high altitudes were significantly higher than those at low altitudes (*p* < 0.01). The concentrations of G6Pase and FBPase at high altitudes were very significantly greater compared to those at low altitudes (*p* < 0.05). There was no significant difference in PC between high and low altitudes (*p* > 0.05). The glucose content at high altitudes was significantly higher than that at low altitudes (*p* < 0.01), and the ratio of the glucose content at high altitudes to that at low altitudes was 1:1.13. The results of gluconeogenesis gene determination analyses at different altitudes show (Figure 4) that the relative expressions of *FOXO1* and *G6PC1* at high altitudes were very significantly higher than those at low altitudes (*p* < 0.01), and the relative expression of *PCK2* at high altitudes was significantly higher than that at low altitudes (*p* < 0.05); there was no significant difference in the relative expressions of fructose-bisphosphatase 1 (*FBP1*) and *PPARGC1A* between high and low altitudes (*p* > 0.05).

### 2.5. Rumen Microbiota Density–VFAs–VFAs Transporter Gene Correlation Analysis

There were certain correlations between the density of the rumen microbiota–VFAs–VFAs transport genes (Figure 5). Among them, the bacterial groups related to propionic acid and ammonia nitrogen were the most numerous. Propionic acid was significantly negatively correlated with *R. flavefaciens*, *R. amylophilus* and *Prevotella* (*p* < 0.05), while it was extremely significantly negatively correlated with *R. albus* (*p* < 0.01). The only extremely significant positive correlation was concentrated between the transporter gene *NHE3* and valerate (*p* < 0.01). Compared with the other bacterial groups, the *R. flavefaciens* and *R. albus* bacterial groups were closely related to most VFAs, among which *R. flavefaciens* was significantly negatively correlated with acetic acid, propionic acid, isobutyric acid and butyric acid (*p* < 0.05), while *R. albus* and VFAs were not significantly correlated; there was no significant correlation found between isobutyric acid, butyric acid and valeric acid (*p* > 0.05), but there was a significant negative correlation between acetic acid, isovaleric acid and ammonia nitrogen (*p* < 0.05). In addition, *R. albus* was significantly correlated with acetic acid (*p* < 0.05). Among the transporter gene–VFAs correlations, the *MCT4* and *AE2* genes were the most significantly correlated with related VFAs. Among them, *MCT4* had a very significant negative correlation with acetic acid and isovaleric acid (*p* < 0.01), a very significant negative correlation with the butyric acid concentration in VFAs (*p* < 0.01) and a significant negative correlation with propionic acid (*p* < 0.05). Except for *AE2*, which was not significantly related to valeric acid (*p* > 0.05), the other VFAs were related to it. 

### 2.6. Correlation Analysis of Rumen VFAs-Hepatic Gluconeogenesis Function

There is a certain correlation between rumen VFAs and liver gluconeogenesis function (Figure 6). Among them, the correlation analysis of the VFAs–key enzyme–glucose contents showed that among the four key enzymes, PC and PEPCK had consistent correlations with VFAs, mainly showing extremely significant negative correlations with acetic acid, propionic acid, isovaleric acid and ammonia nitrogen in VFAs (*p* < 0.01). For the other two enzymes, one of them, FBPase, had significant negative correlations with acetic acid and valerate (*p* < 0.05); the other, G6Pase, had a very significant negative correlation with butyric acid (*p* < 0.01) and a significant negative correlation with isobutyric acid (*p* < 0.05). Among the Glu contents, the correlation between valeric acid and isovaleric acid was not significant (*p* > 0.05). In the analysis of VFAs–gluconeogenesis-related genes, the *PPARGC1A* and *G6PC1* genes were not significantly correlated with VFAs (*p* > 0.05), while *FOXO1* and *PCK2* were significantly negatively correlated with propionic acid (*p* < 0.05), and *FBP1* showed a significant negative correlation with acetate and butyrate. There was a significant negative correlation (*p* < 0.05).

## 3. Discussion

The rumen is the main site for ruminant animals to degrade feed and absorb nutrients. Grass produces metabolites such as VFAs through anaerobic microbial fermentation in the rumen. As an important source of energy, VFAs provide 70–80% of the energy required by ruminants. Among them, acetic acid, propionic acid and butyric acid account for more than 95% [7]. Research has shown that altitude changes can affect the gut microbiota of animals and further affect the structure and function of mammalian gut microbiota [23]. In the process of animal evolution, the host and its gut microbiota have evolved unique physiological mechanisms to adapt to different altitudes. Fan’s study revealed that the abundance of the animal gut microbiota in high-altitude areas is higher than in middle and low-altitude areas [24]. In this study, the relative densities of *R. flavefaciens*, *R. albus*, *F. succinogenes*, *S. ruminantium*, *T. bryantii*, *C. butyricum*, *R. amylophilus* and *B. fibrisolvens* were significantly higher at high altitudes than at low altitudes, indicating that more microbial communities are involved in host gut metabolism in high-altitude areas. In addition, studies have shown that the dominant microbial taxa in the gut of yaks are Firmicutes and Bacteroidetes and that the abundances of these two phyla in high-altitude yaks is higher than those in the low-altitude group [25]. Among them, *R. flavefaciens*, *R. albus*, *C. butyricum*, *R. amylophilus* and *B. fibrisolvens* belong to Firmicutes [26,27]; and *F. succinogenes*, *S. ruminantium* and *T. bryantii* belong to Bacteroidetes [27,28,29,30]. Therefore, the oxygen concentration is lower in high-altitude areas, which promotes the proliferation of anaerobic-tolerant microorganisms, resulting in a higher abundance of the microbial communities compared to animals in low-altitude areas. High-altitude Tibetan sheep are affected by cold stress and show an increased dependence on carbohydrates and require more energy [31]. Additionally, rumen microorganisms convert the indigestible carbohydrates in feed into VFAs such as acetic acid, propionic acid and butyric acid. These VFAs are not only important products of rumen fermentation but are also the main components of energy sources for ruminants. Therefore, with the fermentation provided by microbial communities, the VFAs produced also exhibit adaptive changes. In this study, the proportions of acetic acid, propionic acid, isobutyric acid, butyric acid, isovaleric acid and ammonia nitrogen in the rumen VFAs of the high-altitude group were lower than those in the low-altitude group. The reason for this may be that in high-altitude environments, Tibetan sheep need an energy supply to adapt to cold stress and hypoxic environments. Therefore, the transport capacity of VFAs in Tibetan sheep increases at high altitudes. As the main source of energy, 50–85% of VFAs produced in the rumen are effectively transported and absorbed into the bloodstream by the rumen epithelium through passive diffusion or specific carriers, thereby providing energy for the body [32]. Morphologically, four different cell layers can be distinguished in the surface of the lumen, which is sequentially divided into the stratum corneum, granular layer, spinous cortex and basal layer, from the outside to the inside [33]. There are specific carriers on each layered structure, including *MCTs* that are located on the basal outer side of epithelial cells, *DRAs* that are located on the apical membrane of epithelial cells, *AE2* that is located on the outer membrane of the basement membrane and *NHEs* that are located on the apical and basal outer membranes. Among them, there are two types of *MCTs*, namely *MCT1* and *MCT4*, which mainly transport intracellular SCFA^−^ and their metabolites, namely ketones and lactate [34]. *DRA* mainly exchanges SCFA^−^ and HCO_3_^−^ ions [35]; *AE2* mainly exchanges SCFA^−^ and HCO_3_^−^ ions [36]. There are three types of *NHEs*, *NHE1*, *NHE2* and *NHE3*, which mainly regulate Na transport and the cytoplasmic pH. They are closely related to the absorption of VFAs [37]. Fan’s study revealed that high-altitude yaks may more effectively transport and absorb VFAs than low-altitude yaks, indicating that the transport rates of VFAs in high-altitude Tibetan sheep are greater than those in low-altitude sheep [24]. In this study, the relative expression levels of the VFA transporter-related genes *MCT1*, *MCT4* and *AE2* in the high-altitude group were higher than those in the low-altitude group, indicating that the up-regulation of VFA transporter gene expressions in high-altitude Tibetan sheep improved the VFA transport capacity. Studies have shown that genes related to the transport and absorption of VFAs are significantly up-regulated in the rumen epithelium of high-altitude ruminants, which is consistent with previous studies [20]. Among them, *MCT1* and *MCT4* are monocarboxylic acid transporters that participate in the proton-coupled bidirectional transport of monocarboxylic acids [38]. They are expressed in various tissues, including the rumen epithelium. Moreover, due to its ability to transport intracellular SCFAs^−^ and lactate, it is closely related to energy metabolism in Tibetan sheep. Another transporter gene, *AE2*, mainly participates in the transport of VFAs by exchanging SCFA^−^ and HCO_3_^−^ ions through specific amino acid residues in the transmembrane domain of the rumen epithelium. By analyzing the characteristics of rumen microbial density–VFAs–VFAs transport genes, it was found that there were interactions among the three genes. The three major cellulose-degrading bacteria in the rumen, namely *R. flavefaciens*, *R. albus* and *F. succinogenes*, can efficiently degrade cellulose [26]. The correlation analysis results revealed that the relative densities of *R. flavefaciens*, *R. albus* and *F. succinogenes* were negatively correlated with the propionic acid contents in VFAs, indicating that as the relative density of fiber-degrading bacteria increased, the content of propionic acid and other VFAs in the rumen decreased. As the altitude increases, the biomass and height of aboveground grass decrease, and the crude fiber content of grass decreases [39,40]. Moreover, the dry matter content increases [41]. Therefore, the fiber content in forage is lower at high altitudes than at low altitudes. The strong responsiveness of fiber-degrading bacteria to feed on fiber can affect the production and accumulation of VFAs. As the main VFA-producing bacteria, fiber-degrading bacteria exhibit a decrease in propionic acid content and a negative correlation with cellulose-degrading bacteria. In this study, *R. amylophilus* and *Prevotella* were negatively correlated with propionic acid, as they are starch-degrading bacteria and fiber-degrading bacteria, respectively. It can be inferred that these two types of bacteria are related to the production of propionic acid. Studies have shown that microorganisms of the *Prevotella* genus can ferment and synthesize succinic acid and propionic acid in the intestine [42]. *R. amylophilus* can activate the acrylic ester pathway. With lactic acid as an intermediate, the generation of propionic acid can be promoted [43], which confirms the speculation on the mechanism involved. In addition, the contents of acetic acid, propionic acid and isovaleric acid in VFAs were negatively correlated with *MCT4* and *AE2*, indicating that *MCT4* and *AE2* play crucial roles in the transport of acetic acid, propionic acid and isovaleric acid. VFAs are highly dependent on epithelial surface area and transporter protein expression [9]. *AE2* and *MCT4* serve as specific transporters for VFAs in the rumen epithelium, and they are closely linked to the transport and absorption of VFAs. Therefore, the efficient transport of VFAs leads to increased expression levels of the transport genes *MCT4* and *AE2*, and these increases are negatively correlated with the contents of acetic acid, propionic acid and isovaleric acid in VFAs within the rumen. Previous studies have found that nitrogen-free leachates (soluble carbohydrates) exhibit a slightly lower trend at high altitudes than at low altitudes [41]. Soluble carbohydrates can produce propionic acid through the action of microorganisms. Comprehensive analysis revealed that Tibetan sheep increase the propionic acid content in their VFAs through the action of the fiber-degrading bacteria *R. flavefaciens*, *R. albus*, *F. succinogenes*, *Prevotella* and *R. amylophilus* in their rumen microbiota. To adapt to various environmental pressures such as hypoxia and low temperatures, Tibetan sheep transport acetic acid, propionic acid and isovaleric acid through the rumen epithelial transport genes *MCT4* and *AE2* to the portal vein to participate in liver glycogenesis, resulting in a decrease in volatile fatty acids such as propionic acid in the rumen at high altitudes (Figure 7).

Gluconeogenesis refers to the process by which noncarbohydrate precursors (e.g., propionic acid and lactic acid) produce glucose under the regulation of key gluconeogenic enzymes and genes [18]. Studies have shown that the blood supply to the liver comes from the hepatic portal vein, which is connected to the mesenteric vein and contains metabolites from the digestive tract and microorganisms in venous vessels [44]. Therefore, VFAs can be transported to the hepatic portal vein through transport carriers to participate in gluconeogenesis, and this pathway can be used to supply glucose. In this study, the total amounts of VFAs such as propionic acid at high altitudes were lower than those at low altitudes because the VFAs in high-altitude areas are mediated by the rumen epithelial pathway for gluconeogenesis. At high altitudes, VFAs enhance gluconeogenesis and increase liver glucose contents by up-regulating the interaction among key gluconeogenic enzymes and related genes [45]; this is a mechanism by which the body adapts to low-oxygen environments. In this study, the concentrations of the key enzymes, PEPCK, FBPase and G6Pase, at high altitudes were greater than those at low altitudes, and the glucose contents in the liver at high altitudes were higher than those at low altitudes. At high altitudes, due to other environmental factors such as low oxygen partial pressures and declining grass quality, the body’s energy demand increases, which activates PEPCK to participate in the conversion of pyruvate to phosphoenolpyruvate (PEP) [46]. After phosphorylation, F1,6BP is produced, which is further decomposed into D-fructose 6-dihydrogen phosphate under FBPase catalysis [47] and glucose 6-phosphate (G6P) under the catalysis of fructose isomerase [48]. Finally, G6Pase catalyzes the gluconeogenesis pathway to produce glucose in the endoplasmic reticulum system [49]. This increases the ability of gluconeogenesis to compensate for the energy demand under low-oxygen conditions and improves the host’s adaptability to high-altitude environments. Studies have shown that the enzyme activities of PC, *PCK2*, hexokinase (HK) and G6Pase are all high in the sheep liver [50]. The magnitude of enzyme activity reflects the degree of gluconeogenesis in the body [51,52]. The research results are consistent with those of this study [50]. In addition to the activation of gluconeogenesis-related enzymes, gluconeogenesis-related genes and transcription factors also play crucial roles in gluconeogenesis regulation. Studies have shown that C6Pase is a key enzyme in gluconeogenesis and that *FOXO1* is an important regulatory factor that is essential for regulating the liver gluconeogenesis pathway. Activated *FOXO1* activates C6Pase transcription by binding to the G6Pase promoter, which promotes gluconeogenesis [40,53]. In this study, the relative expression levels of the *FOXO1* and *G6PC1* genes were significantly higher at high altitudes than at low altitudes, indicating that *FOXO1*, as a forked transcription factor, promoted the expression of the gluconeogenic gene *G6PC1*. A study revealed that the use of a recombinant adenovirus-encoding *FOXO1* to transduce renal cells leads to insulin-induced inhibition of dex/cAMP-induced G6p expression [54]. Our research results coincide with their findings. In addition, compared to those in normoxic environments, *PCK2* expressions are up-regulated in low-oxygen environments [55]. Since high-altitude Tibetan sheep are exposed to hypoxic and low-temperature environments, the relative expression level of *PCK2* in the high-altitude group was significantly higher than that in the low-altitude group. At high altitudes, the host’s energy demand is higher than that at low altitudes, so transcription factors and gene expressions are up-regulated during gluconeogenesis, leading to an increase in glucose content, and thus the need for energy. In summary, the expression characteristics of key VFA–gluconeogenesis enzyme–gluconeogenesis genes were analyzed, and it was found that there were interactive relationships among the three genes. VFAs are the main source of energy, among which propionic acid is the main precursor of gluconeogenesis in the liver of ruminants [52]. It is utilized in the liver of Tibetan sheep to produce glucose. In this study, acetic acid, propionic acid, isobutyric acid and isovaleric acid were negatively correlated with PC and PEPCK at different altitudes. Among them, propionic acid was negatively correlated with the *FOXO1* and *PCK2* genes. It can be inferred that the transport genes *MCT4* and *AE2* can transport most of the VFAs to the hepatic portal vein for gluconeogenesis. The contents of acetic acid, propionic acid, isobutyric acid, isovaleric acid and ammonia nitrogen in the rumen at high altitudes are lower than those at low altitudes. Gluconeogenesis is the main way of producing glucose, and propionic acid is the main precursor of gluconeogenesis, which can provide more than 80% of the glucose required by Tibetan sheep. The concentrations of key enzymes involved in gluconeogenesis and the relative expression levels of genes determine the glucose content in the liver. Therefore, the enhanced activity of key enzymes in the gluconeogenesis pathway and the up-regulation of gene expression led to an increase in glucose content and a negative correlation. Comprehensive analysis revealed that the increase in the efficiency of utilizing acetic acid, propionic acid, isobutyric acid, isovaleric acid and ammonia nitrogen by Tibetan sheep at high altitudes enables them to adapt to high-altitude environmental pressures by increasing the activity of key enzymes, including PC and PEPCK, and the expression of related genes during gluconeogenesis in the liver (Figure 7).

## 4. Materials and Methods

### 4.1. Experimental Design and Sample Collection

Twelve Tibetan sheep (n = 6/group) were randomly selected from Zhuoni (low altitude, 2500 m; Gansu Province) and Yushu (high altitude, 4500 m; Qinghai Province). The selected individuals were 3.5 years old, not pregnant, had similar body weights, were in good health and were managed by natural grazing without any supplemental feeding.

When the sheep returned from grazing, the rumen fluid was collected on the next morning using a gastric tube rumen sampler for subsequent VFA determination and DNA extraction. Subsequently, slaughter sampling was performed in accordance with the requirements of the ethics committee and used local traditional slaughtering and sampling methods. First, the rumen tissue was removed and 2 × 2 cm pieces were cut. After the residual rumen fluid was rinsed with precooled saline at 4 °C, the rumen epithelium was isolated with blunt tweezers for subsequent RNA extraction. Second, the liver tissue was cut into small pieces and placed into cryovials for subsequent RNA extraction, enzyme activity and glucose content determination. The above samples were frozen in liquid nitrogen and stored in a freezer at −80 °C after being transported to the laboratory. 

### 4.2. Determination of Rumen VFAs and NH_3_-N Content

The determination of rumen VFAs was carried out using a Shimadzu gas chromatograph (GC-2010 plus, Kyoto, Japan). First, the rumen fluid was centrifuged at 5000× *g* for 10 min, 1 mL of the supernatant was transferred to a 1.5-mL centrifuge tube, 0.2 mL of 25% metaphospacid solution containing 2 EB of the internal standard was added, the mixture was thoroughly mixed and was placed in an ice box for 35 min. After removal, the supernatant was centrifuged at 10,000× *g* for 10 min, and the supernatant was removed using a disposable needle tube. Then, an organic phase filter was used to filter the liquid into a 2-mL brown sample bottle. It was stored at −20 °C for subsequent machine measurements. For the mobile phase composition and flow rate, nitrogen was used as the carrier gas; the flow rate was optimized according to the column specifications, and the flow rate was adjusted according to peak shape and separation effect. The column was an AT-FFAP (30 m × 0.32 mm × 0.25 μm). The chromatographic conditions were as follows: the inlet temperature was 250 °C and the detector temperature was 260 °C. The column temperature was maintained at 60 °C for 1 min, increased to 115 °C at 5 °C/min, and then further increased to 180 °C at 15 °C/min. The concentration of fluid ammonia nitrogen in the rumen was determined by the method proposed by Yang [22].

### 4.3. Determination of Liver Enzyme Activity and Glucose Content

ELISA kits were used to determine liver enzyme activities (catalog numbers: cj962686, cj332625, cj312658 and cj232688). The liver glucose content was measured using a biochemical reagent kit, catalog number: cj332671. The test kits were obtained from Shanghai Enzymes Biotechnology Co., Ltd. (Shanghai, China) The experimental process strictly followed the instructions that were supplied with the reagent kit. The instrument used for the measurements was a spectrophotometer (Thermo 3020, Thermo Fisher Scientific China Co., Ltd., Waltham, MA, USA). In an Excel 2021 worksheet, the standard concentrations were used as the x-axis, and the corresponding OD values of the samples were used as the y-axis to construct the standard linear regression curve. The concentration and content of each liver sample were calculated according to the curve equation.

### 4.4. Total RNA Extraction and cDNA Synthesis from Rumen Epithelial and Liver Tissues

Total RNA was extracted from the rumen epithelium and liver tissue using TRIzol reagent (Invitrogen, Thermo Fisher Scientific, Waltham, MA, USA). To prepare the consumables, enzyme-free gun heads, centrifuge tubes, PCR tubes and other consumables were used, and the scissors and tweezers were autoclaved and dried in a drying oven for reserve use. To prevent RNA degradation, extraction was carried out in an ice box and an ultraclean workbench. The extraction consumables corresponding to each sample were not cross-used. For RNA extraction, 60 mg of the sample was removed from the −80 °C freezer and placed in a 1.5-mL centrifuge tube. Then, 3 magnetic beads were added to each tube, and 1000 μL of lysis solution was injected. The centrifuge tube was placed in a high-speed, low-temperature tissue grinder (KZ-III-FP, Servicebio Technology Co., Ltd., Wuhan, China) that was precooled to −10 °C in advance and ground for 2 cycles (5 times per cycle). Afterward, the sample was removed and centrifuged at 4 °C for 5 min using a frozen centrifuge (Sigma-Aldrich, 1-14K, St. Louis, MO, USA) at 12,000× *g*/min. The supernatant was pipetted into a second centrifuge tube, and a 1/5 volume of chloroform was added. After slight shaking and mixing by vortexing, the mixture was placed in an ice box for 5 min and centrifuged again for 5 min at 12,000× *g*/min using a frozen centrifuge. With 400 μL of the upper water phase absorbed into another 1.5-mL centrifuge tube, 400 μL of isopropyl alcohol was added, mixed upside down, left to stand in an ice box for 10 min and then centrifuged for 10 min. When a white precipitate was observed at the bottom of the tube, the supernatant was discarded, and the precipitate was retained. After adding 1000 μL to the precipitate, 75% ethanol was used to float the precipitate for 3 min, and the precipitate was centrifuged again with a freezing centrifuge at 12,000× *g*/min for 5 min. Then, the centrifuge tube was opened and allowed to dry naturally for 3 min. After the addition of 60 μL of dissolved DEPC, the precipitate was stored at −80 °C until the integrity and absorbance (OD) values were measured. After the detection of RNA integrity by agarose gel electrophoresis, 2.5 μL was taken to detect the OD value twice using an ultra-microspectrophotometer (Thermal Nano Drop-2000, Thermo Fisher Scientific, Waltham, MA, USA), once per μL. The OD values (A260/A280) ranged from 1.8 to 2.1. For cDNA synthesis, the ToloScript RTEasyMix (Tolo Bio, Wuhan, China) for qPCR (with 2-Step gDNA Erase Out) reverse transcription kit was used for cDNA synthesis, and the reverse transcription process was carried out strictly according to the instructions provided with the kit. The concentration (ng) and absorbance (OD) values were measured after reverse transcription, and the samples were diluted with DEPC water according to the concentration and stored at −20 °C for RT-qPCR analyses.

### 4.5. Total DNA Extraction and Dilution of Rumen Microbiota

The DNA of the rumen fluid was extracted using a soil fecal DNA extraction kit (catalog no: TD601-50, Beijing Jianshi Biotechnology Co., Ltd., Beijing, China). The extraction process was carried out strictly in accordance with the instructions provided with the kit. Other steps and precautions were carried out according to the manufacturer’s instructions for total RNA extraction. The OD values (A260/A280) ranged from 1.6 to 1.8, and the concentration and purity of each sample were recorded. Based on the concentration, the solution was diluted 3 times proportionally and stored at −80 °C for subsequent microflora quantification.

### 4.6. Primer Design and RT-qPCR Amplification

With β-actin as the internal reference gene, and bacteria as the internal reference for the bacterial community, the NCBI BLAST website (https://www.ncbi.nlm.nih.gov/tools/primer-blast) (accessed on 18 December 2023) was used for primer design. Primer-specific tests were performed using the Nucleotide BLAST (https://blast.ncbi.nlm.nih.gov/Last.cgi) (accessed on 20 December 2023) website and DNAMAN software (Version 6.0). The primers used were synthesized by Beijing Aoke Dingsheng Biotechnology Co., Ltd. (Beijing, China). The sequences of the gluconeogenesis genes, transport genes and the bacterial population density are detailed in Table 2, Table 3 and Table 4. The RT-qPCR total system had 20-μL wells per well (10 μL of 2 × Q3 SYBR qPCR, 0.4 μL of each of the upstream and downstream primers, 7.2 μL ddH_2_O and 2 μL of cDNA template). For the quantitative reaction parameters, predenaturation was performed at 95 °C for 30 s. Cyclic reactions (40 cycles) occurred at 95 °C for 10 s and 60 °C for 30 s. Dissolution curves were drawn (95 °C for 15 s, 60 °C for 60 s and 95 °C for 15 s).

### 4.7. Statistical Analysis

The experimental data were organized in Excel 2021. All the quantitative results were analyzed with the 2^−ΔΔCT^ method, Ct values were calculated, and the final experimental data are presented as the mean ± standard deviation (SD). Independent sample *t*-tests were performed via SPSS 22.0 software. *p* < 0.05 was considered statistically significant, *p* < 0.01 was considered extremely significant, *p* < 0.05 was considered significant and *p* > 0.05 was considered not significant. After using the Spearman method for correlation analysis, the data were visualized using GraphPad Prism 9.5 and Origin 2021 software.

## 5. Conclusions

Research has shown that there are differences in the densities of rumen microbiota of Tibetan sheep at different altitudes. With increasing altitude, the density of the rumen microbiota increases. The synergistic effect of the VFA transporter-related genes *MCT4* and *AE2* decreases the rumen VFA content in high-altitude (4500 m) Tibetan sheep, but the expressions of key gluconeogenic enzymes and related genes are up-regulated, increasing the glucose content to maintain body homeostasis. Therefore, Tibetan sheep regulate their high-altitude adaptability through interactions between the rumen microbiota and liver axis.

## Figures and Tables

**Figure 1 ijms-25-06726-f001:**
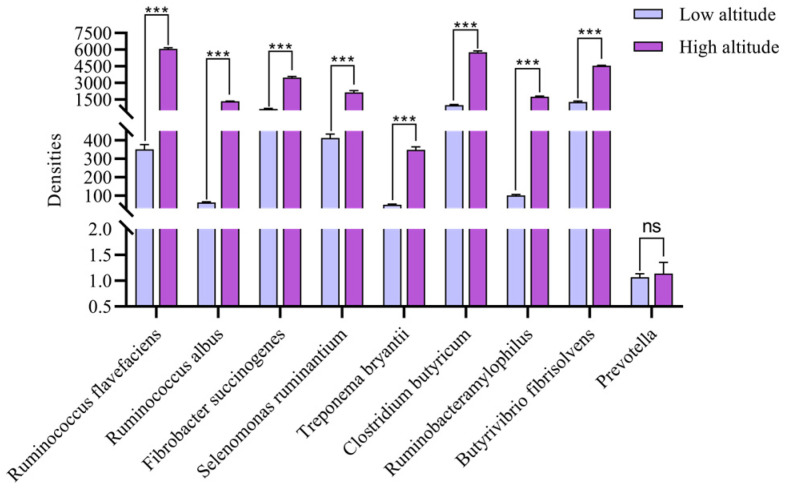
Rumen microbiota densities at different altitudes. Note: *** indicates a highly significant difference (*p* < 0.001); ns indicates that the difference is not significant.

**Figure 2 ijms-25-06726-f002:**
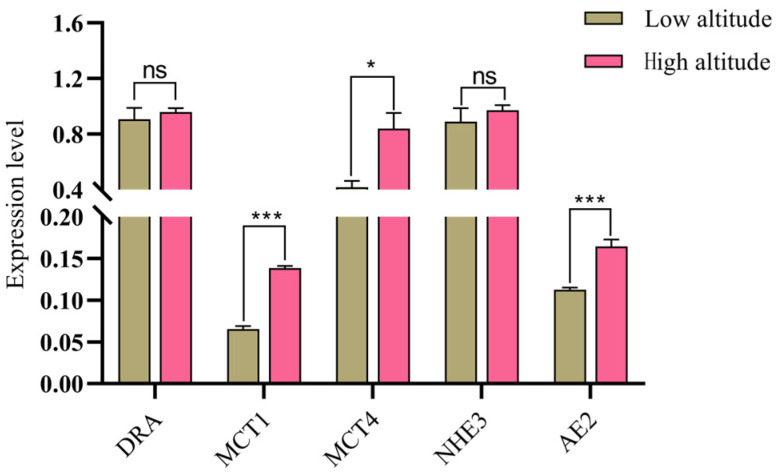
Expression of rumen epithelial transporter genes at different altitudes. Note: *** indicates a highly significant difference (*p* < 0.001); * Indicates a significant difference (*p* < 0.05); ns indicates that the difference is not significant.

**Figure 3 ijms-25-06726-f003:**
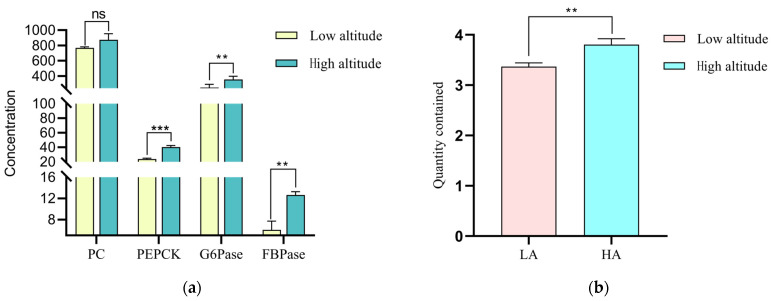
Liver gluconeogenesis key enzyme content and glucose content in Tibetan sheep at different altitudes: (**a**) Liver key enzyme content at different altitudes; (**b**) liver glucose content at different altitudes. Note: *** indicates a highly significant difference (*p* < 0.001); ** indicates a highly significant difference (*p* < 0.01); ns indicates a non-significant difference.

**Figure 4 ijms-25-06726-f004:**
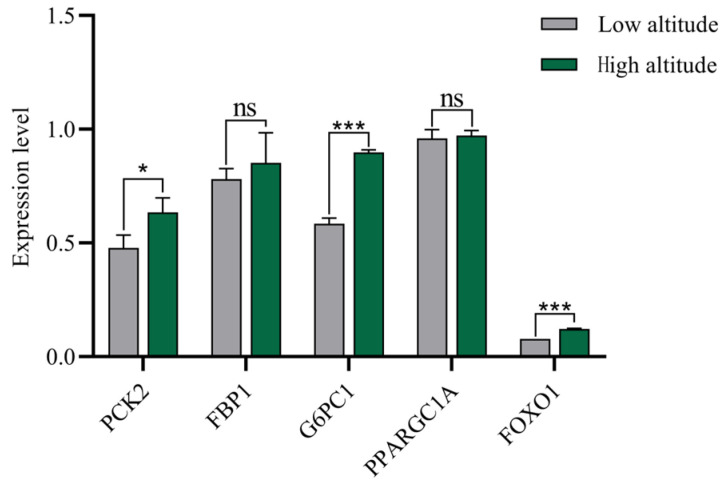
Expressions of liver gluconeogenesis-related genes at different altitudes. Note: *** indicates a highly significant difference (*p* < 0.001); * indicates a significant difference (*p* < 0.05); ns indicates that the difference is not significant.

**Figure 5 ijms-25-06726-f005:**
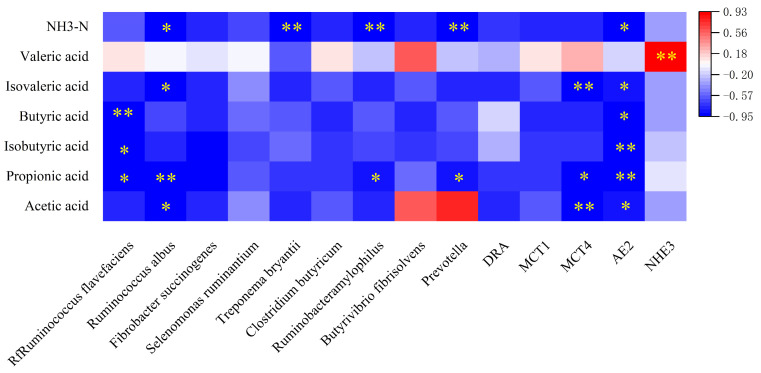
Rumen microbiota density–VFAs–VFAs transporter gene correlations heat map. Note: Correlation heatmap * *p* < 0.05, ** *p* < 0.01.

**Figure 6 ijms-25-06726-f006:**
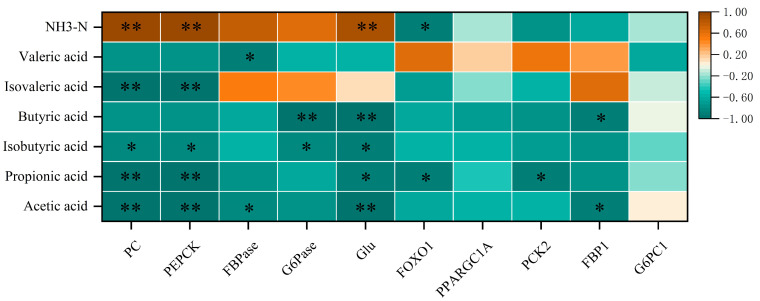
Heat map of rumen VFAs–hepatic gluconeogenesis function correlations. Note: Correlation heatmap * *p* < 0.05, ** *p* < 0.01.

**Figure 7 ijms-25-06726-f007:**
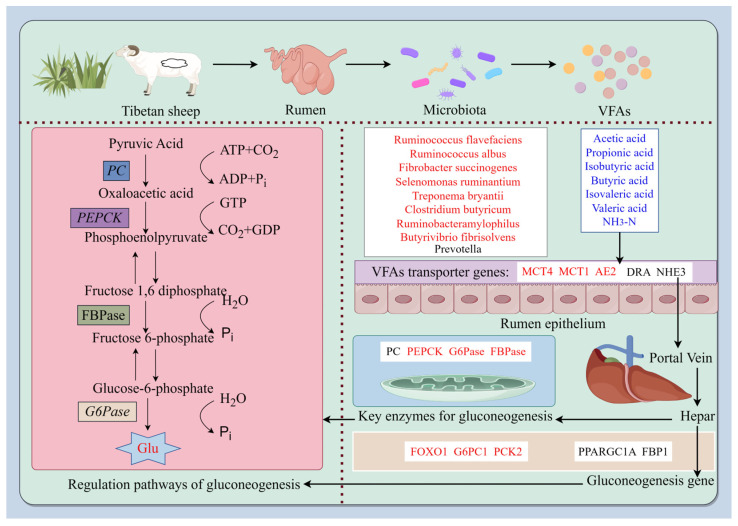
The interaction model of Tibetan sheep rumen microbiota density–VFAs–liver gluconeogenesis axis. Note: The bold black solid line with an arrow in the figure represents the mechanism path of the rumen microbiota density–VFAs–hepatic gluconeogenesis axis. In terms of expression, red fonts represent up-regulation, blue fonts represent down-regulation, and black fonts represent insignificant differences.

**Table 1 ijms-25-06726-t001:** Rumen fermentation parameters at different altitudes.

Fermentation Parameters	Low Altitude	High Altitude	*p*-Value
Acetic acid/(mmol/100 mol)	52.75 ± 2.16	34.04 ± 1.45	<0.001
Propionic acid/(mmol/100 mol)	7.48 ± 0.21	7.01 ± 0.10	0.002
Isobutyric acid/(mmol/100 mol)	1.43 ± 0.06	0.73 ± 0.03	<0.001
Butyric acid/(mmol/100 mol)	7.25 ± 0.23	3.46 ± 0.16	<0.001
Isovaleric acid/(mmol/100 mol)	1.97 ± 0.08	0.73 ± 0.02	<0.001
Valeric acid/(mmol/100 mol)	0.23 ± 0.02	0.23 ± 0.02	0.732
Total amount/(mmol/100 mol)	71.10 ± 2.40	46.19 ± 1.51	<0.001
NH_3_-N/(mg/100 mL)	10.10 ± 0.09	5.46 ± 0.04	<0.001

Note: The concentration difference is extremely significant (*p* < 0.001); the concentration difference is significant (*p* < 0.05); the concentration difference is not significant (*p* > 0.05).

**Table 2 ijms-25-06726-t002:** Primer sequences for detecting the expression levels of liver gluconeogenesis-related genes.

Gene	Primer (5′–3′)	Length/bp	Tm/°C	Login ID
*β-actin*	F: AGCCTTCCTTCCTGGGCATGGA	113	60	NM_001009784.3
	R: GGACAGCACCGTGTTGGCGTAGA			
*FOXO1*	F: GTTGCCCAACCAAAGCTTCC	98	60	XM_027973596.2
	R: TTTAAGTGTAGCCTGCTCGC			
*PCK2*	F: GCGGCTGAACACAAAGGGAA	84	60	XM_015096868.3
	R: AAGGTAGCGCCCAAAGTTGT			
*FBP1*	F: CCAGCTGCTCAACTCGCTTT	90	60	XM_004004092.5
	R: CCAGCTATTCCATAGAGGTGCG			
*G6PC1*	F: GCGGCTGAACACAAAGGGAA	135	60	XM_012186137.4
	R: AAGGTAGCGCCCAAAGTTGT			
*PPARGC1A*	F: GATTGGCGTCATTCAGGAGC	84	60	XM_004009738.5
	R: CCAGAGCAGCACACTCGAT			

**Table 3 ijms-25-06726-t003:** Primer sequences for detecting the expression levels of VFAs transporter-related genes.

Gene	Primer (5′–3′)	Length/bp	Tm/°C	Login ID
*β-actin*	F: AGCCTTCCTTCCTGGGCATGGA	113	60	NM_001009784.3
	R: GGACAGCACCGTGTTGGCGTAGA			
*DRA*	F: CTCCAACAACACCCCGAACA	179	60	NP_001009254.1
	R: ACTACCTCACAGTGTCTGCC			
*MCT1*	F: GCCACCACCAGTGAAGTGTC	138	60	CAC86965.1
	R: ACTGCCTGATAAGATGCCACC			
*MCT4*	F: GTTGGGGATGGATGGTCGTA	185	60	XP_042108082.1
	R: CCACCAGCAACAAGGAAAGC			
*AE2*	F: GTGACGGTACCCGGCTTT	141	60	XP_042105196.1
	R: GTCTTCCTCCCCATAGCTGC			
*NHE3*	F: GAGTCCTTCAAGTCCGCCAA	100	60	XM_042233997.1
	R: GAATGCTGCTGTTTCTCCGC			

**Table 4 ijms-25-06726-t004:** Primer sequences for rumen microbiota density detection.

Nucleotide	Primer (5′–3′)	Length/bp	Tm/°C	Login ID
*Bacteria*	F: CCTACGGGAGGCAGCAG	181	60	*
	R: TTACCGCGGCTGCTGG			
*Ruminococcus flavefaciens*	F: CTAATCAGACGCGAGCCCAT	196	60	LT976286.1
	R: ACATGCAAGTCGAACGGAGT			
*Ruminococcus albus*	F: GGGCTTAACCCCTGAACTGC	114	60	X85098.1
	R: TCGCCACTGATGTTCCTCCT			
*Fibrobacter succinogenes*	F: GATGAGCTTGCGTCCGATT	110	60	EU606019.1
	R: ATTCCCTACTGCTGCCTCC			
*Selenomonas ruminantium*	F: TCTTTCGAGCTGTTGTCCCC	137	60	LT976403.1
	R: GGCGTGCTTAACACATGCAA			
*Treponema bryantii*	F: GCGGTAAGATTGGTGCTTGC	74	60	NR_118718.2
	R: CACAGAGGTACGTCACCCAC			
*Clostridium butyricum*	F: CATTGGGACTGAGACACGGC	108	60	NR_042144.1
	R: AAGACCGTCATCACTCACGC			
*Ruminobacter amylophilus*	F: GGGGACAACACCTGGAAACG	124	60	Y15992.1
	R: CTTGGTAGGCCGTTACCCCA			
*Butyrivibrio fibrisolvens*	F: GGTGAGTAACGCGTGGGTAA	132	60	NR_025981.1
	R: GACGCGGGTCCATCTCATAC			
*Prevotella*	F: AACGCGTATCCAACCTTCCC R: ATTCCACGTCGGATGTCGTC	93	60	NR_181266.1

Note: * Indicates that the bacteria are housekeeping gene sequences (16S rRNA sequences).

## Data Availability

The original contributions presented in the study are included in the article; further inquiries can be directed to the corresponding author/s.

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
