# Peer review of "Effects of the Interaction between Rumen Microbiota Density–VFAs–Hepatic Gluconeogenesis on the Adaptability of Tibetan Sheep to Plateau"

_ijms, 2024, doi:10.3390/ijms25126726_

Round 1
Reviewer 1 Report
Comments and Suggestions for Authors
In this study, Yang et al. investigated how the density of rumen bacteria and the production of volatile fatty acids (VFAs) in Tibetan sheep at different altitudes influence hepatic gluconeogenesis, revealing adaptations to high-altitude environments. This study presents a meaningful topic, but it has some issues. Here are some comments on this study:
1. Line 28 “The conclusion shows” I consider it should be “the results showed”.
2. Please define the abbreviation such as “VFAs” when it is first used in abstract, results, and methods.
3. Method section follows the result section, Table 4 should be Table 1. Standard deviation and mean need to have the same reserved digits after the decimal point.
4. It is recommended that the authors revise and improve the description of the results in sections 2.5 and 2.6; it is not necessary to present each result with the same expression.
5. Line 453 “60 g of the sample”, please make sure that 60 g of sample was used to extract the RNA.
6. Line 478 “fluorescence quantification” should be RT-qPCR. Line 495 “ddH2O 7.2 μL” and Line 497 “95 ℃ for 10 s and 60 ℃”, ℃ is different from the others.
7. Line 277 “SCFA- and HCO3- ions” please check it.
Comments on the Quality of English LanguageIt is necessary to polish the English language
Reviewer 2 Report
Comments and Suggestions for Authors
This is an interesting study on high-altitude Tibetan sheep exposed to environments such as hypoxia and low temperature, and the impact of this environment on the relative expression level of key digestive enzyme systems and rumen function as compared to their counterparts in low-altitude area.The quality of presentation is good and just a few minor corrections are recommended.
Convert VFA to standard units-mmol/100 mol or % as molar proportion. This is the standard format.
L302-308: When fibre decreases with high altitude, what proportion of feed dry matter increases? Are forages in hih altitude high in soluble carbohydrates- tending towards more propionate? See L323-324
Reference should follow a consistent pattern. See number 10,16 in capital letters.
Round 2
Reviewer 1 Report
Comments and Suggestions for Authors
I thank the authors for addressing all my comments.